

# Evaluation of In-situ observations on Marine Weather Observer during the Typhoon Sinlaku

Wenying He[1,2], Hongbin Chen[1,2], Hongyong Yu[3], Jun Li[1], Jidong Pan[1], Shuqing Ma[4],
Xuefen Zhang[4], Rang Guo[4], Bingke, Zhao[5], Xi Chen[6], Xiangao Xia[1,2], Kaicun Wang[7]

[1]*Key Laboratory of Middle Atmosphere and Global Environment Observation, Institute of Atmospheric Physic, Chinese Academy of Sciences, Beijing 100029, China*

[2]*School of the Earth Science, Chinese Academy of Science University, Beijing 100049, China*

[3]*State Key Laboratory of Earth Surface Processes and Resource Ecology, College of Global Change and Earth System Science, Beijing Normal University, Beijing, China*

[4]*Meteorological Observation Center of the China Meteorological Administration, Beijing 10081, China*

[5]*Shanghai Typhoon Institute of CMA, Shanghai   200030, China;*

[6]*Shanghai Marine Meteorology Center, Shanghai Meteorology Center, Shanghai 200030, China;*

[7]*Peking University, Beijing 100029, China*

**Correspondence**: Wenying He (hwy@mail.iap.ac.cn) and Hongbin Chen (chb@mail.iap.ac.cn)

## Abstract

The mobile ocean weather observation system, named Marine Weather Observer (MWO), developed by the Institute of Atmospheric Physics (IAP), consists of a fully solar-powered, unmanned vehicle and meteorological and hydrological instruments. One of the MWOs completed a long-term continuous observation, actively approaching the Typhoon Sinlaku center from July 24 to August 2, 2020, over the South China Sea. The in-situ and high temporal resolution(1-min) observations obtained from MWO were analyzed and evaluated by comparing with the observations made by two types of buoys during the evolution of Typhoon Sinlaku. First, the air pressure and wind speed measured by MWO are in good agreement with those measured by the buoys before the typhoon, reflecting the equivalent measurement capabilities of the two methods under normal sea conditions. The sea surface



temperature (SST) between MWO and the mooring buoys is highly consistent

throughout the observation period and even less difference after the typhoon's arrival,

indicating the high stability and accuracy of SST measurements from MWO during the

typhoon evolution. The air temperature and relative humidity measured by MWO have

significant diurnal variations, generally lower than those measured by the buoys,

which may be related to the mounting height of the sensor. When actively approaching

the typhoon center, the air pressure from MWO can reflect some drastic and subtle

changes, such as a sudden drop to 980 hPa, which is difficult to obtain by other

observation methods. As a mobile meteorological and oceanographic observation

station, MWO has shown its unique advantages over traditional observation methods,

and the results preliminary demonstrate the reliable observation capability of MWO in

this paper.

**1 Introduction**

Marine meteorological hazards, including typhoons, fog, strong winds, and many

other extreme weather events, occur frequently over China (Xu et al., 2009). In

particular, typhoons that make landfall off the southeast coast of China cause direct

economic losses of about 0.4% of gross domestic product and more than 500 deaths

per year (Lei, 2020). Many efforts have been made in recent decades to improve the

understanding of typhoon genesis and evolution and the forecasting of typhoon paths

(Bender et al. 2007; Black et al. 2007; Sanford et al. 2007; Bell et al. 2012). However,

errors in model initial conditions remain the main cause of typhoon forecast



uncertainty due to the scarcity of real-time ocean meteorological observations, especially in distant waters (Zheng et al. 2008; Rogers et al. 2013; Emanuel and Center 2018). Currently, marine observations over China are very limited and rarely occur in the deep ocean (Dai et al., 2014). This situation greatly limits the development of marine meteorology, especially the improvement of typhoon forecasting. Therefore, there is a urgent need to develop advanced observation techniques at sea. With the rapid development of satellite communication and navigation technology as well as sensor technologies in recent years, marine unmanned autonomous observation systems have been increasingly broken and applied at sea (Lenan and Melville, 2014; Wynn et al., 2014; Thomson and Girton, 2017).

To obtain more meteorological observations at sea, the Institute of Atmospheric Physics (IAP), Chinese Academy of Sciences, has developed an automatic and mobile marine weather observations system based on a solar-powered, unmanned vehicle, named Marine Weather Observer (MWO). To test the observation capability and endurance, one of the MWOs cruised over the South China Sea from June to August 2020, during which a tropical cyclone formed and turned into a weak typhoon. The MWO was then remotely controlled to actively approach the center of Typhoon Sinlaku on August 1st, 2020, providing valuable in-situ observations for typhoon research and forecasting (Chen et al., 2021, hereafter Chen21).

To better understand the quality of observations obtained from MWO, we directly compared the observations of MWO and several buoys around it over the South China



Sea during the evolutions of Typhoon Sinlaku. The outline of the paper is described
below. In Section 2, we briefly describe Typhoon Sinlaku and the observations
obtained from MWO and the buoys. Then MWO observations and the comparisons
with buoys observations are presented in Section 3. The observation difference
between MWO and buoys are discussed in Section 4, and finally a summary is given
in Section 5.

**2 Typhoon Sinlaku and the related observations**

Typhoon Sinlaku (No. 2003) formed as a tropical depression over the South
China Sea on July 31, 2020, then intensified into a typhoon on August 1. The center of
the typhoon crossed Hainan Island, China at a speed of 25 km/h and finally made
landfall off the coast of Thanh Hoa City, Vietnam, at 0840 UTC on August 2.
To better monitor the evolution of Typhoon Sinlaku, MWO was used for the first
time to obtain in-situ meteorological observations under extreme sea conditions. The
detailed MWO design and performance were described in Chen21. Measurements of
atmospheric and oceanic environment variables are accomplished with instruments
mounted on MWO, including the AirMar 220WX automatic weather station, mini-CT
sensor, and pyranometer. High temporal resolution (1 minute) data on atmospheric
temperature and humidity, air pressure, wind speed, wind direction, sea surface
temperature (SST), seawater conductivity, and total radiation can be automatically
transmitted to the ground control center via the Beidou communication satellite.
Detailed technical specifications of the meteorological and hydrological sensors can
be found in Chen21.
To evaluate the quality of the observations obtained from MWO, we mainly
compared them in this paper with the buoy observations conducted simultaneously
during the typhoon Sinlaku observation experiments from July 22 to August 4 (Zhang
et al., 2021,Qin et al., 2022). The buoy data consisted mainly of five mooring and two
drifting buoys that were able to provide the same environmental variables measured
on MWO from July 23 to August 2 but with a 10-minute interval. Thus, the 1-minute
observations from the MWO were averaged into 10-minute results and then matched
with the 10-minute observations from the buoys. More than 1300 matched samples at
10-minute intervals were obtained from July 24 to August 2, 2020, covering the main
evolution periods of Typhoon Sinlaku in the South China Sea.

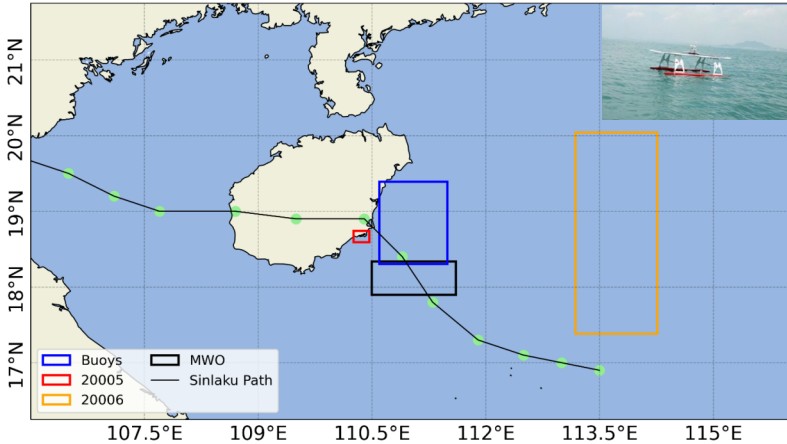


Fig.1. Observation ranges of three observation methods, including 5 mooring buoys in the blue
box, 2 drifting buoys (20005 and 20006), and MWO(as shown in the small photo in the upper right
corner). The red, orange, and black boxes are the observation ranges of two drifting buoys and MWO
from July 24 to Aug.2, 2020, respectively. The light green dots on the black line are the locations of
Typhoon Sinlaku during the period from 0600UTC on July 31 to 0000 UTC on August 2, with a
3-hour interval.





From the locations and the observation ranges of the buoys and MWO in Fig.1, it
can be seen that for the two drifting buoys (20005 and 20006, named D05 and D06,
respectively), the drifting range of D05 is very close to the moving area of MWO,
while the drifting path of D06 is about 3-4 degree from MWO in longitude. For the
five mooring buoys in the blue box, one buoy named M64 is the closest, while the
others are located within about 100 km from MWO.

**3 Results**

**3.1 The observations from MWO**

First, Fig.2 presents the time series of environmental variables measured by
MWO at **1-minute** interval from July 24 to August 2, 2020. It can be seen that in the
first stage before the arrival of the typhoon, such as July 24-29, the air temperature and
humidity show a clear diurnal variation and negative correlations, and the air pressure,
SST, and seawater conductivity also show small and stable variation.
Then from late July 29 to August 1, the typhoon moved toward the observation
area of MWO. The wind gradually strengthened, and the wind direction frequently
changed from south to north. The air pressure, air temperature, SST, and seawater
conductivity gradually decreased. On July 31, MWO was about 30 km away from
Typhoon Sinlaku and then actively moved to the predicted path of Sinlaku by remote
control. The drastic changes in air pressure and wind speed can be seen around noon
on August 1st. Unfortunately, the humidity sensor stopped working on July 31.
MWO arrived at the predicted passing area of Sinlaku on August 1st at 0928 LST



(Local Standard Time), with a pressure of 1011 hPa at that time. Then the air pressure
decreased to 992 hPa around 1140 LST and even rapidly dropped to the lowest 980
hPa at 1158 LST. Subsequently, the pressure gradually rose and increased to 992 hPa
at 1256 LST, accompanied by strong winds of 15.1 m/s.

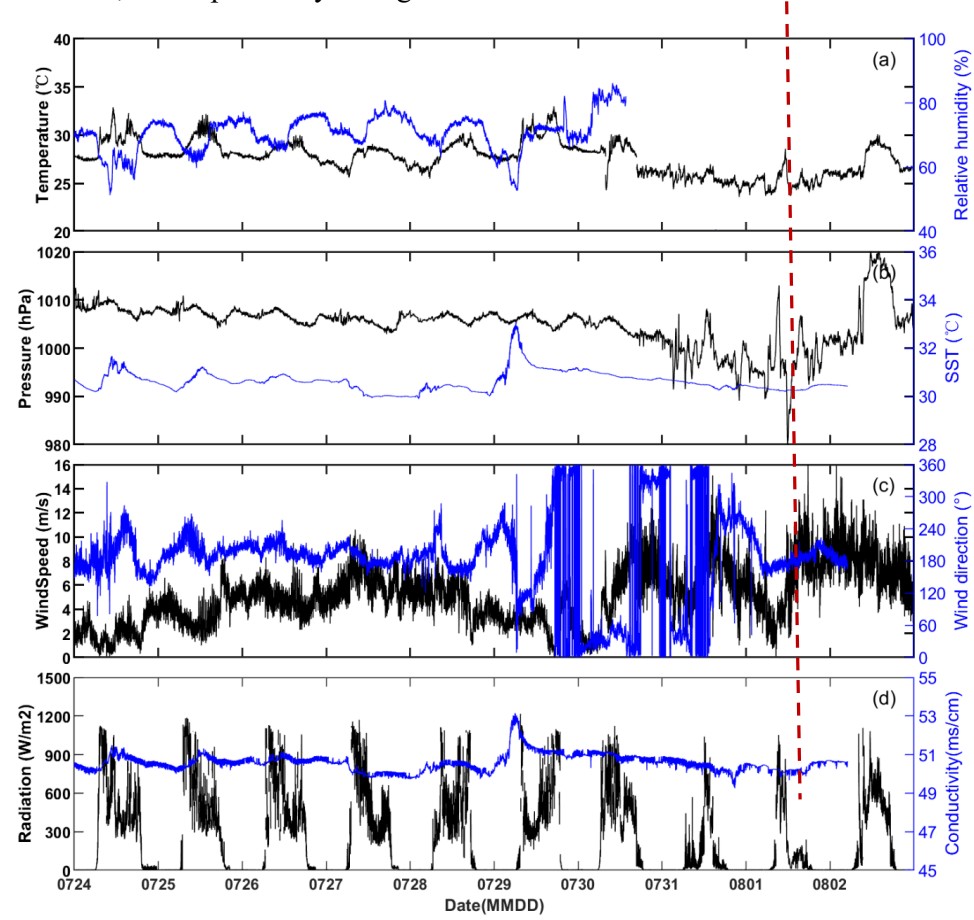


**Fig.2.** Time series (LST) of (a) air temperature and relative humidity, (b) SST and atmospheric
pressure, (c) wind speed and direction, and (d) total radiation and seawater conductivity collected
onboard MWO in the **1-min interval** during the South China Sea typhoon observation experiment
from July 24 to August 02, 2020. The dashed red line represents the nearest times of MWO passing
through the typhoon center.

148        Such drastic fluctuations of air pressure over sea indicated that MWO might be

cross the typhoon center around 1200 LST. The subsequent path verification also


proved that MWO was nearly 2.4 km away from the typhoon path issued by the
Central Meteorological Observatory (CMO) of the China Meteorological
Administration, which reflected that MWO successfully passed through the center of
Typhoon Sinlaku. When Sinlaku was moved away from MWO observation range on
August 2, the wind speed gradually decreased and varied less in direction. Compared
with the normal sea conditions in the first stage, we call the next four days (from July
30 to Aug.2) as the second stage with larger changes in sea conditions.

157        To match the 10-minute observations from the buoy, we reprocessed the 1-minute

observations provided by MWO to the 10-minute average. Usually, under stable sea
conditions, the differences in meteorological variables over time may be slight in the
short term. When the typhoon arrived on August 1 and MWO approached the typhoon
center, the variables measured on MWO showed significant changes in Fig. 2.
Therefore, the difference between 1-minute and 10-minute averaged meteorological
variables may be useful for detecting fine-scale structure during typhoons.

164        Thus, the differences between the 1-minute and 10-minute results for the three

variables, including wind speed, air pressure, and air temperature on August 1 are
shown in Fig.3. It is clear that the trends in air pressure (Fig.3b) are consistent for both
time windows, for example, there are two peaks from 0600 LST to 1000 LST and a
sharp drop to 980 hPa around 1200 LST. the air temperature in Fig.3c also shows a
highly consistent variation in the 1-min and 10-min results. However, there is a
significant difference in the wind speed between the two time windows (Fig. 3a).
Before 1200 LST, both wind speeds are close to each other and are relatively





consistent. As the MWO approaches the typhoon center after 1200 LST, the 1-minute
wind speed varies more significantly than the 10-minute wind speed until 1800 LST. it
is assumed that the 10-minute window may reflect the average state of the wind field
to some extent. the significant difference between the 1-minute and 10-minute wind
speeds reflects the changes in the fine-scale structure of the wind field during the
typhoon evolution. As shown in Fig. 3d, the differences in pressure and temperature in
the two time windows were mostly close to zero and did not vary much throughout the
day on August 1. In contrast, the wind speed varies greatly with different time interval
during most of the day, especially around 0600 LST and 1200-1800 LST, where the
wind speed difference is as high as 5 m/s. This also reflects the apparent fluctuating
behavior of the 1-minute wind field, indicating strong turbulent activity in the
near-surface atmosphere. There has been a lot of research work on horizontal roll and
tornado-scale vortices of typhoons, which are closely related to the drastic changes in
the wind field (Morrison et al. 2005; Lorsolo et al. 2008; Wurman and Kosiba 2018;
Wu et al. 2020). Most of the previous work has been based mainly on landfalling
hurricanes observed by Doppler radar deployed near the coast. In this work, in situ
observations of MWOs that can actively cross typhoon centers in distant oceanic
regions will provide a new perspective to study the fine structural changes during
typhoon evolution.

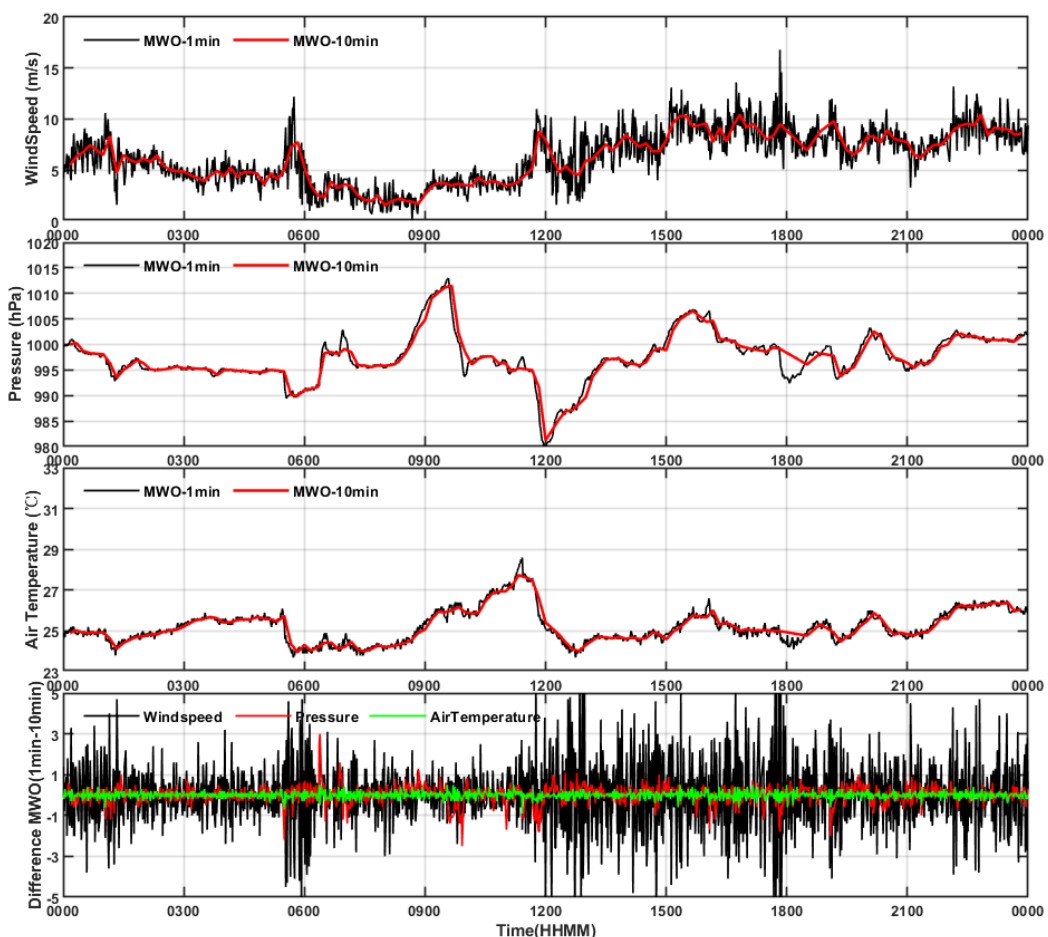

Fig.3 The difference between 1-min and 10-min results for wind speed, air pressure, and temperature on Aug.1.

## 3.2 Comparisons of the observations between MWO and buoys

To assess the quality of MWO observations, we first compared the air pressure and wind speed measured by MWO and all buoys (drifting and moored) as shown in Fig. 4. Before seeing the differences in the observations, it is best to know the spatial distance variation between MWO and the buoys as shown in Fig. 4c. For the two drifting buoys, the D05 was always closer to the MWO, within 100 km, from July 24



to August 2. While D06 gradually moved away from MWO over time, from less than
100 km on July 24 to 400 km on August 2. For the five mooring buoys, M64 is less
than 50 km from MWO from July 24 to 31 and very close to MWO from August 1 to 2.
The rest of the buoys are within 100 km from MWO.
Then for the air pressure comparison in Fig. 4a, all buoys and the MWO
measurements in the first stage match very well and basically overlap, except for a
slight difference in the farthest D06. With the arrival of the typhoon, the measured
pressure from MWO changed more obviously, especially around 1200 LST on August
1 the lowest pressure was about 980 hPa when MWO was close to the typhoon center.
In addition, an abnormally high pressure was measured on MWO around 14:00 on
August 2, and the cause of the abnormality is unknown at present. The pressure
measured by the buoys was relatively close and consistent throughout the period,
except for a slight change in the farthest buoy D06.
The wind speeds measured from buoys and MWO (Fig.4b) have a good
consistency. They are very close to each other in the first stage due to stable sea
conditions, especially the closer buoys D05 and M64. In the second stage, especially
from July 31 to August 1, there are enhanced changes in wind speed due to the passing
of the typhoon. In the first half of August 1st, there was a significant trend difference
in wind speed from MWO and buoys, for example, the former gradually decreased and
reached its minimum value when MWO is closing to the typhoon center about 1200
LST, while the latter mostly increased during this period. Subsequently, in the second
half of August 1st, the wind speed from MWO rapidly increases to 10m/s, more



consistent with those measured from buoys and almost superimposed. As the typhoon
gradually moved away from the observation domain of MWO and buoys on Aug.2, all
wind speeds became closer and gradually decreased, returning to the first stage state.

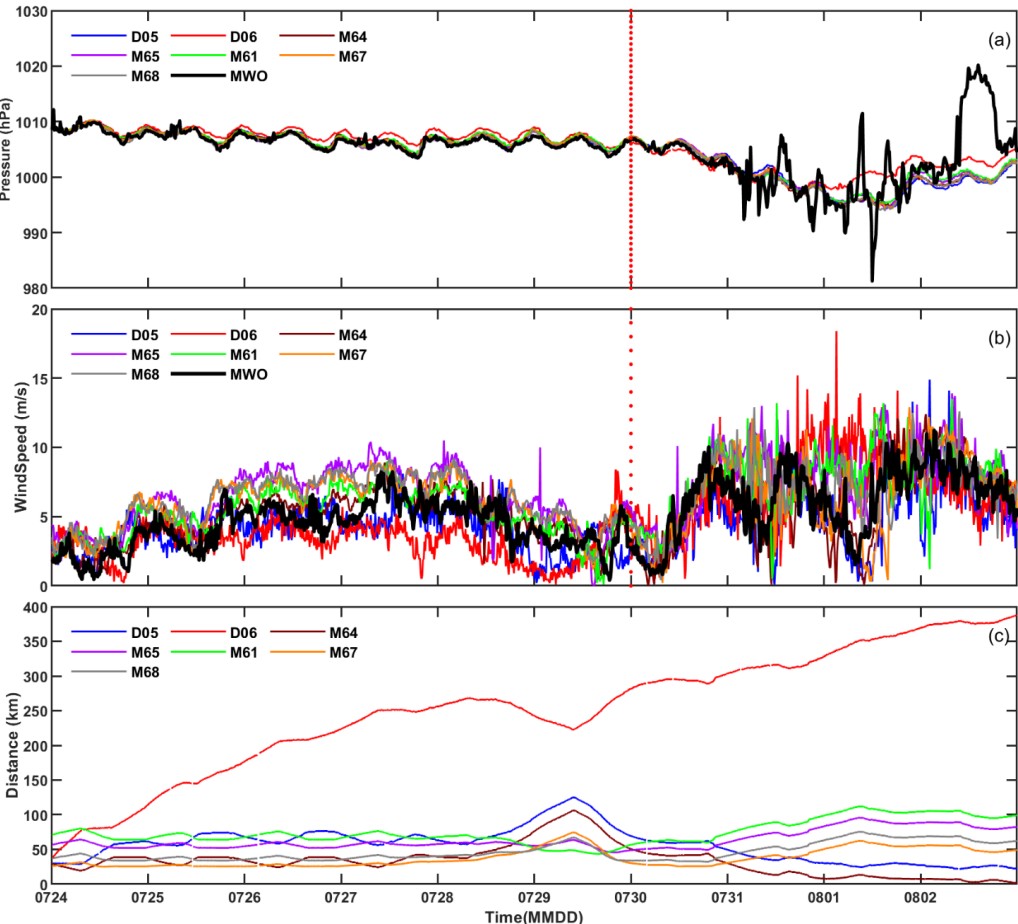


**Fig.4.** Time series (LST) of (a) air pressure and (b) wind speed collected from seven buoys (2
drifting and 5 mooring, legend begin with D and M, respectively) and MWO from July 24 to August
02, 2020. The dashed red line is on July 30 to separate the first and second stages.

Similarly, air temperature and SST obtained from MWO and buoys are compared
in Fig.5. It seems in Fig.5a that air temperature from MWO is generally lower than
those from buoys most of the time, especially during the night of the first stage and



when approaching the center of the typhoon in the second stage. The diurnal variations
of air temperature measured from MWO and the drifting buoy D05 are more
significant and close in the first stage. Relatively, the air temperature differences
among the mooring buoys are smaller and more stable in the first stage, then enhanced
due to the coming of the typhoon.
For SST shown in Fig.5b, the observations from MWO during the entire period
are very close to those from the five mooring buoys, and are more consistent,   even
showing peak areas simultaneously, except for the slight difference from July 27-29.
For the two drifting buoys, the SST measured by the D05 buoy is 1-2 ℃ lower than
that measured by MWO on July 27-30, while SST measured by the D06 buoy is more
stable and close to that measured by MWO.
In addition, seawater conductivity and relative humidity (RH) can be obtained
from MWO. However, only the two drifting buoys can provide seawater conductivity
measurement,   and   the   mooring   buoys   can   provide   relative   humidity   (RH)
measurement. Hence, the seawater conductivity and RH measured from MWO are
compared with those from the corresponding available buoys and displayed in Fig.5c.
Firstly, the seawater conductivity measured on MWO and two drifting buoys are
very different, but the detailed values of each instrument are constant throughout the
entire period. The conductivity measurement from D06 buoy is the highest, generally
exceeding 60 mscm$^{-1}$, followed by D05 buoy, which is basically around 57 mscm$^{-1}$,
and the lowest is about 50 mscm$^{-1}$ from MWO.
The RH difference between mooring buoys and MWO shown in Fig.5c is only
available in the first stage because the humidity sensor on MWO stopped working
after July 30. The RH variations are similar to those of air temperature, that is, RH
from MWO is mostly lower than that from mooring buoy, especially in the daytime.
The diurnal variations of RH measured from MWO are more significant while RH
differences among the mooring buoys are smaller and stable in the first stage.

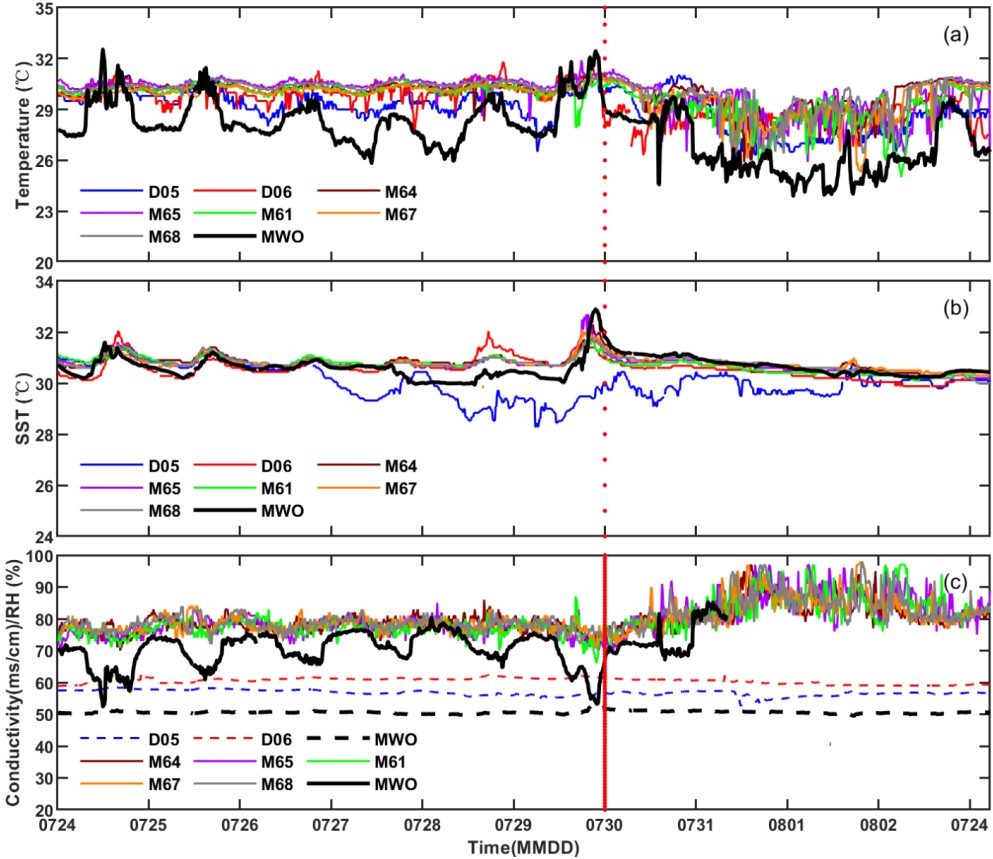


**Fig.5.** Same as Fig.3, except for (a) air temperature, (b) SST, and (c) seawater conductivity (dotted
line) for drifting buoys and RH (solid line) for the mooring buoys.

To better see the influence of typhoon moving on MWO observations, Fig.6
shows the scattering plots of meteorological variables observed by MWO and the





nearest buoys, including the drifting D05 and the mooring M94. The color samples
and their corresponding statistical results are used to quantify the observations
differences before (in red) and after the arrival of typhoons (in blue). Firstly, before the
arrival of the typhoon, air pressure differences between MWO and both buoys are in
good agreement, as shown in the red samples in Fig.6a,b. Both air pressure differences
are very close and smaller, such as mean bias error (MBE) and standard deviation
(STD) less than 0.5 hPa. However, in the second stage, the pressure difference is
significantly enhanced when MWO approaches the center of the typhoon, shown as
the highly scattered blue samples in Fig. 6a, b, with corresponding STD up to 3.5 hPa.
The wind speed measurements from both buoys and MWO have good
consistency in both stages, which is reflected in the good overlap of the red and blue
samples in Fig.6c,d, and the corresponding MBE and STD are very close. For SST
shown in Fig.6e,f, it is seen that the observations between MWO and the mooring
M64 buoy are quite consistent with a difference of less than 0.3℃ before and after the
coming of the typhoon. The SST measurements from the drifting buoy D05 are more
scattering with those from MWO the most of time, especially significantly decreased
by about 1-2 ℃ from July 27 to Aug. 1st as shown in Fig.5b. The overall MBE and
STD of SST difference are less than 1.0 ℃ due to partial overlap of the samples.

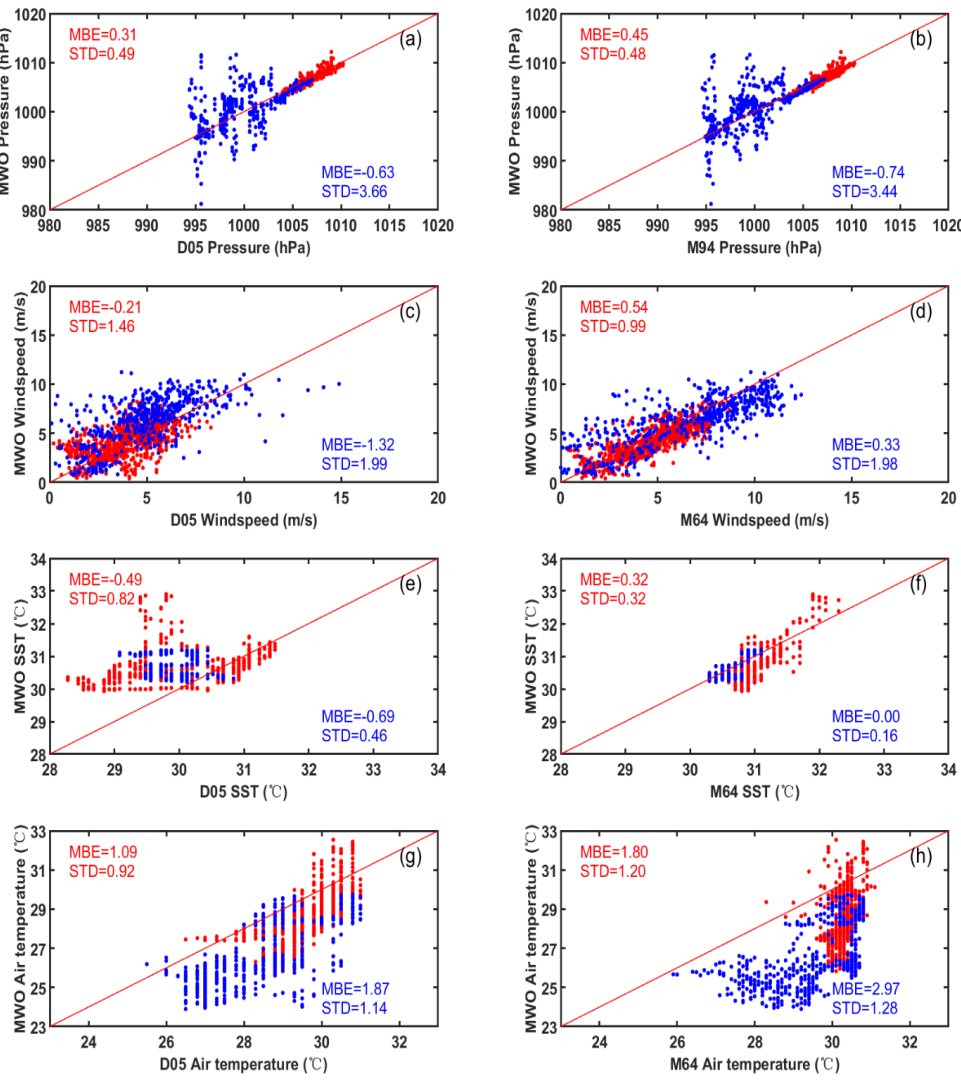


Fig.6 Scattering plots of observations from the nearest buoys and MWO, with the drifted D05 in
the left column and the mooring M64 in the right column. From top to bottom, they are air pressure,
wind speed, SST, and air temperature, respectively.
Regarding air temperature, the observations from MWO show significant
fluctuations, while the mooring M64 shown in Fig.6h mostly fixes around 30℃ in the
first stage. In the second stage, the air temperature measured from MWO is lower than



that measured from both buoys, for example, the MBE corresponding to buoys D05
and M64 is close to 1.9℃ and 3 ℃, respectively. Relatively, the changed trends of air
temperature measured from MWO and D05 have good consistency in both stages.

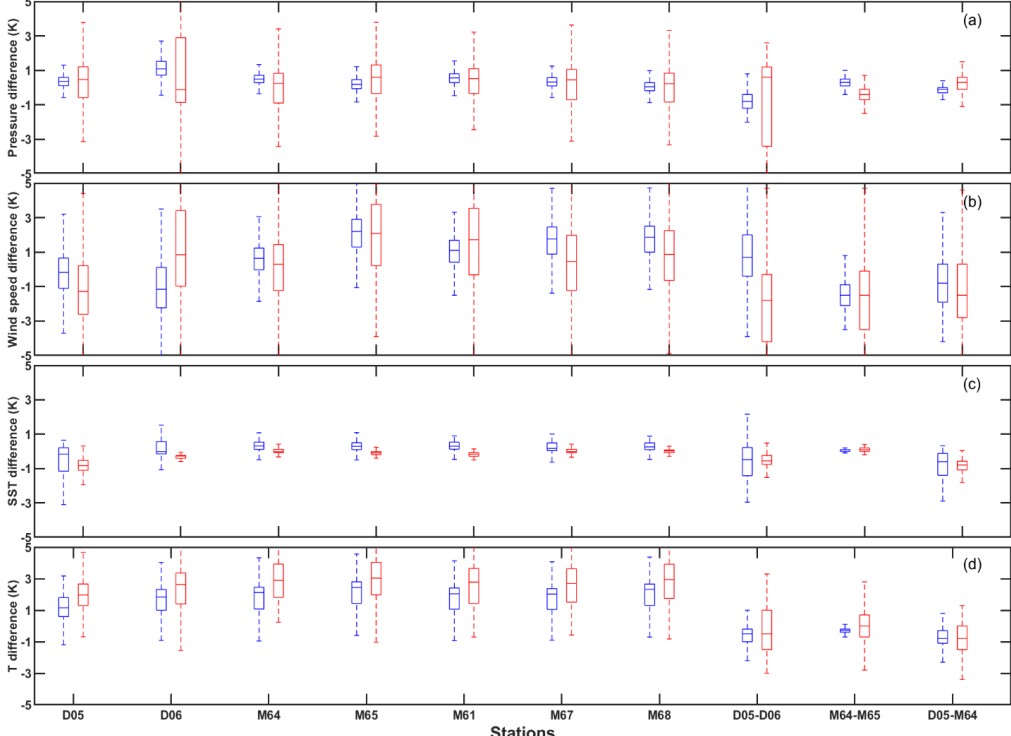


Fig.7. The boxplots of observations difference (blue: the first stage; red: the second stage) between
MWO and seven buoys, as well as between buoys (i.e. D05 and D06, M64 and M65, and D05 and
M64). The observations from up to bottom are air pressure (a), wind speed(b), SST (c), and air
temperature (d).

To better understand the observed differences between MWO and buoys, as well
as between buoys, the boxplots in Fig. 7 show the distribution of their differences in
pressure, wind speed, SST, and air temperature during the first (blue) and second (red)
stages. The center marker in each box indicates the median, and the bottom and top
edges of the box indicate the 25th and 75th percentiles, respectively. The first seven





buoys reflect the difference between the buoy observations and MWO observations.
The last three reflect differences in observations between buoys, including the two
drifting buoys D05 and D06, the nearest (M64) and farthest mooring buoys(M65)
from the MWO, and the nearest drifting D05 and moored M64 from the MWO.
The pressure difference in Fig. 7a shows a clear change in the first and second
stage. Before the arrival of the typhoon, the pressure difference between MWO and
the buoys are close to zero, and the magnitude of the differences between MWO and
the buoys vary relatively uniformly, indicating that the pressure measured by MWO
has the same level of accuracy as thoset measured by buoys under normal sea
conditions. In the second stage, the range of pressure difference between MWO and
buoy is 2-3 times larger than that in the first stage, but the median value of pressure
difference is still relatively close, mostly within 1hPa. Relatively, the pressure
differences between the buoys in both stages are relatively small and stable, except for
the farthest D06.
The median difference of wind speed between MWO and the buoys are mostly
within 1 m/s as shown in Fig. 7b. The wind speed difference in the second stage is
significantly larger than that in the first stage. The wind speed difference between
buoys seems to increase with the distance between buoys, as in the more distant buoys
D06 and M65. In general, the wind speed differences between MWO and buoys are
comparable to the wind speed differences between buoys.
For the SST in Fig. 7c, the observed differences between MWO and the moored
buoys are very small throughout the period and even better in the second stage. In



contrast, the difference in SST between MWO and the two drifting buoys is not as
good as that between the moored buoys, especially for the closest buoy, D05, which
fluctuates more in the first period, which may indicate that the SST quality of D05
buoy is not as good as its other measurements, such as pressure and wind speed.
The difference in air temperature between MWO and the buoys (Fig. 7d) is more
pronounced than the difference in SST. Because of the lower temperature measured by
MWO, the median of temperature difference with the buoys is mostly positive, e.g., 1
K in the first stage and 2 K in the second stage, while the temperature difference
between the buoys is smaller in the first stage and increases significantly by a factor of
2-3 in the second stage.
**4 Discussions**
In this paper, we first used 1-minute MWO in-situ observation data to monitor the
changes in air pressure, wind field, temperature, and humidity before and after the
arrival of typhoons. In particular, the air pressure significantly decreased from 1010
hPa under normal sea conditions to 980 hPa at the time when MWO crossed the center
of the typhoon. During this period the air pressure underwent obvious and detailed
fluctuations, which cannot be provided by previous observations. In addition, the wind
field reflected the detailed and obvious fluctuations when the typhoon approached.
The air temperature and relative humidity in the lower layers of the sea exhibited
obvious diurnal variations. In contrast, SST is more stable, showing slight changes
before and after the typhoon.
Further comparison with buoys observations during the same period revealed that
under normal sea conditions before the arrival of the typhoon, the air pressure and
wind speed measured by MWO and buoys showed good consistency, especially the
difference in air pressure was only less than 0.5hPa, and the wind speed difference was
less than 0.5 m/s. Moreover, the difference between MWO and buoys was comparable
to that of multiple buoys, indicating that the measurement accuracy of air pressure and
wind speed on MWO was equivalent to that of the buoys under normal sea conditions.
With the arrival of the typhoon, the air pressure measured on MWO fluctuated greatly,
while the corresponding measurements on the buoys were more stable, resulting in a
significant pressure difference between MWO and the buoys. This may mainly be
related to the location where MWO crossed the center of the typhoon. In addition, as
the typhoon departed, the air pressure and temperature measured on MWO showed
abnormally high values around 14:00 on August 2nd, and then returned to normal
range at night, which may be related to unknown external interference.

364        The trend of wind speed change between MWO and the buoys was more

consistent before and after the arrival of the typhoon. When MWO was closest to the
center of the typhoon, the wind speed change between MWO and the buoys was
slightly misaligned.

368        For the air temperature and relative humidity under normal sea conditions,

measurements made by the mooring buoys were relatively constant and little
variations in a day; the corresponding drifting buoys measurements showed slight
diurnal fluctuations; MWO measurements fluctuated significantly from day to night.
This may be related to the mounting height of the sensor. Usually, the sensor on the





mooring buoy can reach up to 10m, on the drifting buoy it may be about 1.5m, and on MWO it is close to 1.2m. The closer the sensor is to the water's surface, the more obvious the impact on the marine environment.

Compared with other variables, the SST variation before and after the typhoon's arrival was weak and appeared relatively stable. In particular, the SST measurements from MWO and the mooring buoys were very close throughout the period, and even better in the second stage. However, the larger difference in SST between MWO and the nearest drifting buoy may be caused by the quality of the SST measurement from the latter.

**5 Summary**

During the typhoon observation experiment in the South China Sea in July-August 2020, MWO completed long-term continuous observations, especially by actively approaching the center of Typhoon Sinlaku in the deep sea. The in-situ meteorological and hydrological observations obtained by MWO were evaluated by comparing them with the observations made by two types of buoys during the evolution of Typhoon Sinlaku. We obtained some preliminary results as follows.

1) Before the arrival of the typhoon, air pressure and wind speed measured by MWO and the buoys were in good agreement, with the difference in air pressure less than 0.5hPa and the difference in wind speed less than 0.5 m/s, indicating that the measurement accuracy of air pressure and wind speed obtained by the two methods is comparable under normal sea conditions.

2) The SST observations of MWO and the mooring buoys show highly consistent



in the entire period, and even a smaller difference in SST after the arrival of the
typhoon, demonstrating the high stability and accuracy of SST measurements from
MWO during the typhoon evolution.
3) The air temperature and relative humidity measured from MWO have obvious
diurnal variations and are generally lower than those from the buoys, which may be
related to the mounting height of the sensor.
4) When actively approaching the typhoon center, the air pressure measured by
MWO can reflect some drastic and subtle changes, such as a sudden drop to 980 hPa,
which is difficult to obtain by other observation methods.
As a mobile meteorological and oceanographic observation station, MWO has
shown its unique advantages over traditional observation methods. Although we only
analyzed and evaluated the in-situ observations obtained in one individual case of
MWO crossing the Typhoon Sinlaku in this paper, the results preliminary demonstrate
the reliable observation capability of MWO. For better monitoring of typhoon systems,
it will be necessary to deploy a meteorological and hydrological observation network
composed of multiple MWOs in the future, which will provide comprehensive in-situ
observations on spatial and temporal scales required for forecasting, warnings, and
research of marine meteorological hazards.
***Acknowledgments.*** This work is supported by the National Natural Science
Foundation of China (Grant No. 41627808), the Key Technologies Research and
Development Program (Grant No. 2018YFC1506401), the Shanghai Typhon Reseach
Foundation (Grant No. TFJJ202101). We wish to express our sincere gratitude to



Beijing Chunyi Aviation Technology Co., Ltd., Hainan Meteorological Service, Wang
Hu, and Wang Chunhua of Qionghai Meteorological Service, and all personnel who
participated in this experiment.

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
