# Peer review of "Evaluation of In-situ observations on Marine Weather Observer during"

_Atmospheric Measurement Techniques, 2023_

## Author Comment (AC1)

- **RC1**: ['Comment on amt-2023-120'](), Anonymous Referee #1, 13 Sep 2023  [reply]()

The paper is based on the in-situ, high temporal resolution (1-min) observations of the typhoon Sinlaku over the South China Sea from July 24 to August 2, 2020, with a solar-powered, mobile ocean weather observation system, abbreviated as MWO for Marine Weather Observer and developed by China Institute of Atmospheric Physics (IAP). The data has been analyzed and compared with the observations made by 7 buoys, which are close to the MWO. The results from the study, especially on the variation features of meteorological parameters when Sinlaku passed over the MWO, are clearly presented by the paper and shows that the observation system is technically fine and the data obtained are useful for ocean study. It is hoped that the paper can be published after proper modification.

Specific comments and suggestions

1. Lines 34-36 and 398-400: "The air temperature …… measured from MWO …… are generally lower than those from the buoys, which may be related to the mounting height of the sensor". And "the sensor on the mooring buoy can reach up to 10m, on the drifting buoy it may be about 1.5m, and on MWO it is close to 1.2m. The closer the sensor is to the water's surface, the more obvious the impact on the marine environment" as shown Lines 372-374, Does it imply that the "lower" is because of the marine water temperature is cooler than air? Otherwise, the air temperature measured by MWO should be higher than those from the buoys because the normal air temperature gradient. Please consider this further and supply a detailed analysis.

   Reply: Thanks for your comments. We did check the air temperature and SST measured by MWO. As shown Fig.1 here, the diurnal variation of air temperature were significant before typhoon is coming, especially much lower than SST at night. In Fig.5a, similar lower value at night was found for drifting buoys measurements. We also checked more about drifting buoys. As mentioned in Cao et al. (2019) that the air temperature sensor is installed above the 40 cm buoys ball as shown in Fig.2, which means that it may be located about 1 m above the sea surface, close to the mounted height of the sensor on MWO. Therefore, the air temperature obtained from MWO and drifting buoys displayed similar diurnal variations as shown in Fig.5a, and both were lower than SST, which is consistent with the normal air temperature gradient.

   We have corrected the description regarding this point as "This may be related to the installation height and sensitivity of the sensor. Usually, the

sensor on the mooring buoy can reach up to 10m, on the drifting buoy and MWO it may be about 1.0m (Cao et al., 2019). The closer the sensor is to the surface, the more pronounced the impact of near-surface environmental changes." in line 373-376. We also changed the related descriptions in the paper.

[Figure]

Fig.1 The time series of air temperature and SST measured from MWO from July 24 to August 02, 2020.

[Figure]

Fig.2 Schematic diagram of drifting buoy (cited from Cao et al., 2019,Typhoon observation and analysis of domestic marine meteorological drift buoy experiment,

Meteor Mon, 45(10):1457-143(in Chinese))

2. Please consider if the paragraphs in Section 4 can be concentrated and inserted into Section 3 and

    Reply: Thank you for your suggestion. Considering the overall structure of the paper as well as the different emphasis in the two sections, we had better retain them in order to better understand the relevant results.

3. line 142: The position for"LST"in the caption for Fig. 2 should be adjusted.

Reply: Thank you. We removed the LST in the caption for Fig.2 and Fig.4, and added "It should note that the time used in the following is local time (shorted for LT), also known as Beijing time." In line 125-126.

4. Lines 148-149: The date should be included in addition to the time "1200LST".

Reply: Yes, we changed "1200LST" into "1200 LT on Aug.1" in line 151.

5. Lines 227-229: The subgraph Fig.4c is not mentioned in the caption for Fig.4.

Reply: Thank you. We changed it as "**Fig.4.** Time series of (a) air pressure and (b) wind speed (c) distance for the seven buoys (2 drifting and 5 mooring, legend begin with D and M, respectively) and MWO from July 24 to August 02, 2020. The dashed red line is on July 30 to separate the first and second stages." in line 229-231.

6. Please consider the difference between Drifted buoys and drifting buoys.

Reply: Thank you. All "drifted" words are changed into "drifting".

---

## Author Comment (AC2)

- **RC2**: 'Comment on amt-2023-120', Jiagen Li, 27 Sep 2023  reply

The scarcity of in-situ observational data during typhoon passage has been a significant obstacle to studying the interaction between the ocean and typhoons, especially high-resolution observational methods, which are crucial but limited in supply. The MWO is a mobile observation system driven by all-solar energy and includes meteorological and hydrological observation instruments. This study shows the continuous observations near the center of Typhoon "Sinlaku" in the South China Sea and results show that the MWO has equivalent measurement capability, and the observed air pressure and wind speed are consistent with those measured by buoys. This is a meaningful observational study, but there are some questions that need to be discussed further, as follows:

Minor comments:

1. Figure 1: It is recommended to mark the date and typhoon intensity category on the typhoon track.

Reply: Thanks for your suggestions. We added the date and surface level pressure (SLP) on the typhoon track in Fig.1 as shown here.

[Figure]

Fig.1. Observation ranges of three observation methods, including 5 mooring buoys in the blue box, 2 drifting buoys (20005 and 20006), and MWO(as shown in the small photo in the upper right corner). The red, orange, and black boxes are the observation ranges of two drifting buoys and MWO from July 24 to Aug.2, 2020, respectively. The light green dots marked with date and surface level pressure on the black line are the locations of Typhoon Sinlaku from 0000UTC on July 31 to 0000UTC on August 2, which is from the best track typhoon provided by JMA.

2. Line 119: The relative positions of the five mooring buoys might be marked in Figure 1.

Reply: Thank you. The specific location of the fixed buoy used in this paper is not convenient to disclose at present, so the spatial coverage of those buoy is shown here in a box. Thanks for your understanding.

3. Figure 2: SST, wind direction and seawater conductivity are not drawn to the end, is there no data?

Reply: Sorry for that. That's my mistake. The updated Figure 2 is shown here and in the revised version. Except for RH, which is only available before July,31, the measurement of other variables covers the entire period.

[Figure]

Fig.2. Time series of (a) air temperature and relative humidity, (b) SST and atmospheric pressure, (c) wind speed and direction, and (d) total radiation and seawater conductivity collected onboard MWO in the 1-min interval during the South China Sea typhoon observation experiment from July 24 to August 02, 2020. The dashed red line represents the nearest times of MWO passing through the typhoon center.

4. Lines 175-177 and Figure 3: The difference between 1-min and 10-min averaged wind speed is significant. The best track typhoon wind speed of JMA and JTWC is 1-min and 10-min respectively. Can you simply analyze the difference or advantages of wind speed of these two datasets based on the results of this study?

Reply: Thanks for your suggestions. Fig.3 is the comparison of in-situ measurements in different time resolution. Indeed, the difference of wind speed is significant, and we deduced that it might reflect the apparent fluctuating behavior of the 1-minute wind field, indicating strong turbulent activity in the near-surface atmosphere. For the related model data, such as reanalysis data, we did the comparison of wind speed between ERA5 (red) and MWO (black) as shown here. The varied trends of both data are more consistent throughout the entire period, though the fluctuating range of observed wind are more obvious. I hope this supplement might help to understand your suggestions.

[Figure]

5. Figure 4: The lines are too dense, resulting there are some interference information, there is no need to show such a long time, each stage (stage 1 or 2) could be showed for four days.

Reply: Sorry for such dense lines in Fig.4. The main reason is that there are multiple observation data (7 buoys and 1 MWO) with high temporal resolution, and their measurements are close or fluctuating within a similar range, resulting in most of them overlapping, no matter ten or four days. On the other hand, we are more focused on those matching observations from July 24 to August 2, 2020. To see the overall changes of all the observations within the 10 days, it is best to display them together within a complete time period.

6. Line 227: Figure 4 (c) caption is missing.

Reply: Thank you. We changed it as "**Fig.4.** Time series of (a) air pressure and (b) wind speed (c) distance for the seven buoys (2 drifting and 5 mooring, legend begin with D and M, respectively) and MWO from July 24 to August 02, 2020. The dashed red line is on July 30 to separate the first and second stages." in line 229-231.

7. Figure 6 (b): x-axis label should be 'M64'

   Reply: Thank you. We corrected the wrong name in Fig.6(b), and the updated Fig.6 as shown here.

[Figure]

Fig.6 Scattering plots of observations from the nearest buoys and MWO, with the drifting D05 in the left column and the mooring M64 in the right column. From top to bottom, they are air pressure, wind speed, SST, and air temperature, respectively.

8. Figure 7: The unit of y-axis coordinate is wrong. Besides, it is recommended to add a zero-value line with a dotted line.

Reply: Thanks for your suggestions. We corrected the wrong unit of y-axis, and added the zero-value line with a dotted line in Fig.7. The updated Fig.7 is shown here as

[Figure]

Fig.7. The boxplots of observations difference (blue: the first stage; red: the second stage) between MWO and seven buoys, as well as between buoys (i.e., D05 and D06, M64 and M65, D05 and M64). The observations from up to bottom are air pressure (a), wind speed(b), SST (c), and air temperature (d). The dotted line is zero-value line.

Major comments:

Both the abstract and summary note that "the sea surface temperature (SST) of the MWO and the mooring buoy were highly consistent throughout the observation period, and the difference was even smaller after the arrival of the typhoon." But I think the reason why the SST difference is smaller during the typhoon is because the amplitude of the SST itself is smaller. As we all know, SST should change more significant during typhoons, and the SST amplitude of stage 2 should be larger than that of stage 1. However, the observation results in this paper were exactly the opposite, what are the reasons?

Reply: Thanks for your comments. The scattering plot of SST as shown in Fig.6f does reflect that the varied amplitude of SST in stage 2 is smaller than that in stage 1, resulting in a smaller SST difference in stage 2. This may be

related to the solar radiation in both stages. From Fig.1 it can be seen that the air temperature and SST show significant diurnal variation due to solar radiation in stage 1, and in stage 2 both show slight diurnal variation due to decreasing solar radiation caused by more cloud and rainfall during the arriving of the typhoon. In addition, the mixing disturbance of ocean current during typhoon processes may be another aspect impacting on the variation of SST. In this case, the intensity of Typhoon Sinlaku is not very strong, and its impact on SST is not as significant as we thought, at least not reflected from those observations by MWO and other buoys. To avoid such confusing express, the related sentence is changed into "The sea surface temperature (SST) between MWO and the mooring buoys is highly consistent throughout the observation period, indicating the high stability and accuracy of SST measurements from MWO during the typhoon evolution." in line 30-33 and "The SST observations of MWO and the mooring buoys show highly consistent in the entire period, demonstrating the high stability and accuracy of SST measurements from MWO during the typhoon evolution." In line 393-395.

[Figure]

Fig.6f